# Saliva Influence on the Mechanical Properties of Advanced CAD/CAM Composites for Indirect Dental Restorations

**DOI:** 10.3390/polym13050808

**Published:** 2021-03-06

**Authors:** Teresa Palacios, Sandra Tarancón, Cristian Abad, José Ygnacio Pastor

**Affiliations:** 1Department of Geological and Mining Engineering-CIME, Universidad Politécnica de Madrid, 28003 Madrid, Spain; 2Department of Materials Science-CIME, Universidad Politécnica de Madrid, 28040 Madrid, Spain; sandra.tarancon@upm.es (S.T.); jy.pastor@upm.es (J.Y.P.); 3Department of Prosthodontics, Universidad de Cuenca, Cuenca 010103, Ecuador; cabad02@ucm.es

**Keywords:** mechanical behaviour, microstructure and composition, CAD/CAM, resin composite, saliva immersion influence

## Abstract

This study aims to evaluate the microstructural and mechanical properties of three commercial resin-based materials available for computer-aid design and manufacturing (CAD/CAM)-processed indirect dental restoration: Lava^TM^ Ultimate Restorative (LU), 3M ESPE; Brilliant Crios (BC), COLTENE and Cerasmart^TM^ (CS), GC Dental Product. The three types of resin-based composite CAD/CAM materials were physically and mechanically tested under two conditions: directly as received by the manufacturer (AR) and after storage under immersion in artificial saliva (AS) for 30 days. A global approximation to microstructure and mechanical behaviour was evaluated: density, hardness and nanohardness, nanoelastic modulus, flexural strength, fracture toughness, fracture surfaces, and microstructures and fractography. Moreover, their structural and chemical composition using X-ray fluorescence analysis (XRF) and field emission scanning electron microscopy (FESEM) were investigated. As a result, LU exhibited slightly higher mechanical properties, while the decrease of its mechanical performance after immersion in AS was doubled compared to BC and CS. Tests of pristine material showed 13 GPa elastic modulus, 150 MPa flexural strength, 1.0 MPa·m^1/2^ fracture toughness, and 1.0 GPa hardness for LU, 11.4 GPa elastic modulus; 140 MPa flexural strength, 1.1 MPa·m^1/2^ fracture toughness, and 0.8 GPa hardness for BC; and 8.3 GPa elastic modulus, 140 MPa flexural strength, 0.9 MPa·m^1/2^ fracture toughness, and 0.7 GPa hardness for CS. These values were significantly reduced after one month of immersion in saliva. The interpretation of the mechanical results could suggest, in general, a better behaviour of LU compared with the other two despite it having the coarsest microstructure of the three studied materials. The saliva effect in the three materials was critically relevant for clinical use and must be considered when choosing the best solution for the restoration to be used.

## 1. Introduction

Over the last decade, the number of new restorative materials has increased considerably to meet the increasing concern about dental restorations and aesthetics [1]. Thus, modern restorative dentistry is overwhelmed with the high number of available materials in the dental market, and material selection has become a critical decision [2]. Furthermore, new manufacturing technologies, such as assistant computer-aid design and manufacturing technologies (CAD/CAM), has been incorporated as an excellent alternative to traditional time-consuming methods by reducing the fabrication time up to 90% [3,4,5]. Therefore, a comprehensive study covering the various materials used for CAD/CAM systems is now necessary. The fact that these materials are presented in blocks with a particular volume (variable according to their manufacturer) is an added factor for standardizing the tests and obtaining necessarily clinically results.

Therefore, it is necessary to know the materials’ mechanical and microstructural properties to understand their performance along the time under service. Among the various types of dental materials available, this study is merely focused on resin-based composites. Materials that can be specially designed for a wide range of applications can be quickly processed with excellent mechanical and biological properties and low production costs [6,7]. Notwithstanding, they exhibit a very low fracture toughness and an exceptional susceptibility to fracture due to superficial natural flaws or defects introduced during the final tooth-piece machining [8,9]. Alternatively, resin-based composites are known for their relatively easy mechanizing, simple repairing in the oral cavity and less abrasive effect toward opposing dentition [10]. Despite this, there is some concern about possible allergic reactions in dental personnel and patients [11].

So, mechanical properties and their evolution after saliva immersion are crucial to estimate dental materials’ clinical performance. The most common form of providing material resistance in prosthetic dentistry is to perform flexural strength tests [12,13,14,15,16]. Hence, this study focuses on clarifying the mechanical performance of the studied materials in the most accurate manner. This mechanical characterization was performed under two different conditions, with the materials directly provided by manufacturers and after 30 days in saliva immersion. To get this, the materials were analyzed by X-Ray fluorescence (XRF) and Field Emission Scanning Electron Microscopy (FESEM) to obtain the precise composition and distribution of the microstructure. The mechanical properties, including hardness, flexural strength and fracture toughness under the two states, were measured and correlated with microstructure to obtain the micromechanisms of failure. The obtained results were compared with those published by manufacturers (when available), which are sometimes imprecise.

## 2. Materials and Methods

### 2.1. Materials and Specimen Preparation

The resin-based composite materials used in this study: Lava^TM^ Ultimate Restorative (LU), Brilliant Crios (BC), and Cerasmart^TM^ (CS), were selected over the broad range of chairside CAD/CAM commercial materials currently used for indirect dental restorations. According to the technical documentation, extensive information about the composition as disclosed by manufacturers can be found in Table 1.

The starting materials were C14 blocks with rough dimensions 18 × 16 × 18 mm^3^ (Figure 1). Due to their extreme brittleness, they had to be embedded in epoxy resin and cut-off with an Accutom-50 (Struers, Denmark) using a diamond disk under high-flow-water refrigeration to avoid the damage of the specimens due to overheating during cutting in a two-step process. All the preparation and cutting process was performed under standard laboratory conditions of room temperature (22 ± 2 °C) and 50 ± 10% relative humidity. After being embedded, C14 blocks were cut to slices of 1.5 mm thickness and then embedded again and cut to their nominal beam dimensions 1.5 × 1.5 × 17 mm^3^. Finally, they were cleaned in distilled water for 10 min by ultrasounds and thoroughly dried.

The miniaturized beam specimens were used to perform most of the tests except for the TPB tests to determine fracture toughness. In these specimens, notches were introduced using ultra-short laser ablation, the so-called single edge laser-notch beam (SELNB) method [17]. This technique, which was successfully tested in brittle materials [18,19,20,21], produces very sharp notches in the material (Figure 2) in the order of nanometers and can be considered equivalent to a natural flaw. Additionally, the process is performed with high speed, high accuracy, good reproducibility and precision for reliable fracture toughness testing. Therefore, these notches can be considered a real crack and not an artificial notch with a finite notch radius that can influence the measurement of the material’s real fracture toughness. Consequently, this novel method provides an accurate, reliable, and real measure of the fracture toughness, produces high-quality, reproducible notches with no damage on the rear of the notch root radius and a very low dispersion of results.

### 2.2. Ageing Procedure

To analyze the influence of the saliva’s direct contact with the tooth, half of the specimens were immersed and stored in artificial saliva (AS) at room temperature for 30 days. According to the literature, this time is enough to determine the saliva’s overall influence on the mechanical properties, since no relevant mechanical properties were observed after this period [22]. These specimens were tested under the same conditions as received by the manufacturer (AR) state materials. Finally, the results were also compared with the limited and incomplete data provided by the manufacturer (MN) that could be found in the literature.

### 2.3. Microstructure and Fracture Surfaces Characterization

An Auriga Field Emission Scanning Electron Microscope (FESEM) column from Zeiss (Oberkochen, Germany) was used to examine the studied materials’ microstructure and the fracture surfaces after testing. To analyze the materials’ microstructure, some material specimens were embedded in an epoxy resin since their dimensions are too small to be held in the automated polishing system. After their surface grinding and polishing, they were etched with 5% HF during 30 s to reveal the microstructure: i.e., grain boundaries and the higher contrast between constituents. After that, they were coated with Au. After the TPB tests, post-mortem fracture surfaces of samples were examined to analyze morphology and grain topography con correlate the micromechanisms of fracture with the macromechanical properties obtained in the tests.

### 2.4. X-ray Fluorescence Analysis

Typically, precise information about the studied materials’ elemental composition is hard to obtain; however, this could be critical to clarify mechanical test results. To achieve that, a non-destructive technique such as X-ray fluorescence (XRF) spectrometry was used to determine elemental concentration. Tiny samples of each material—previously cleaned with ethanol—were introduced in a Zetium XRF spectrometer from Malvern Panalytical (Egham, Surrey, UK) to identify the crystalline phases and the chemical elements through semi-quantitative analysis.

### 2.5. Density Measurement

The experimental density was measured via Archimedes method with immersion in high purity ethanol at 22 °C using a Mettler Toledo AG245 (Columbus, OH, USA) and a mass with a resolution of 10^−4^ g. At least six samples of each material were measured.

### 2.6. Vickers Tests

To calculate hardness, two methods were used. The first one was by micro-indentation tests at room temperature using a durometer AKASHI MVK-EIII (Tokyo, Japan) equipped with a Vickers indenter and an applied load of 9.8 N for 12 s by following ASTM E92-27 [23]. These tests were performed in the AR and AS samples. Additionally, AR specimens were also tested with an applied load of 3 N to examine the influence of the force in the results.

### 2.7. Nanoindentation Tests

The second method for obtaining hardness was to measure it with nanoindentation tests. These tests were performed at room temperature using a NanoIndenter XP from former MTS Systems Corporation (Oak Ridge, TN, USA) and a standard Berkovich tip calibrated with fused silica under an applied load of 0.25 and 0.5 N. Around 15 tests were performed under load control for each material and condition. Based on the load-displacement data recorded at all times, the average values (with their standard error) of nanohardness and elastic modulus were calculated based on the Oliver and Pharr method [24,25].

Nevertheless, with these tests, it also was possible to obtain the nanoelastic modulus (nE). This data, along with the fracture toughness, was a real property of the material, representing the fundamental bond between atoms.

### 2.8. Three-Point Bending Tests

The materials’ ultimate strength was measured in bending, i.e., flexural strength, which is a reasonable substitute for the tensile test in materials with low toughness and is the most common test to determine strength in dental materials [26]. Although four-point bending (FPB) is usually employed, here, three-point bending (TPB) tests were selected mainly due to the small size of the provided C14 blocks (Figure 1), as commented in Section 2.1, but also because it used a more straightforward test fixtures than FPB reducing, therefore, frictions and bonding points between supports.

This choice also had implications in the results since tests did not fulfil the minimum size requirements for uniaxial flexural strength standards such as ASTM C1161-18 [27], which requires a load span of at least 20 mm, or for polymerizable resin composites such as the dental standard ISO 4049 [28] which also promote the use of 20 mm span. However, ISO 6872:2015 [29] allows a load span of 12 mm, which was the distance used for this research.

Despite this, and to avoid the problems of performing miniaturized non-standard TPB tests on the smooth samples (without a notch), a system to entirely allocate them for each test in a reproducible way was used to obtain the emplacement shown in Figure 3a. Finally, the upper part was subjected to compression stress, the lower area to tensile stress, and the rolls allowed the beam’s free movement.

From the load values obtained from the tests, the strength at the fracture point for a rectangular cross-section was calculated by using the standard material strength formulas [30]:(1)$$\sigma_{f}=\frac{3}{2}\frac{F_{max}L_{s}}{BD^{2}}$$

Testing the miniaturized SELNB specimens (Figure 3b) with the TPB tests and measuring the fracture toughness of the materials in the AR and AS samples were possible. This property reveals the resistance of materials against crack propagation. Therefore, resistance to fracture in the presence of a natural flaw is a real indicator of the material’s resistance to failure.

For each specimen, the maximum applied load was recorded during the test. The initial notch length was measured using a FESEM to calculate the fracture toughness by using the appropriate formula [31].

## 3. Results

### 3.1. Composition and Microstructure

According to the technical data (Table 1), LU and CS are both composite resin materials with 80% and 71% nanoparticles of mainly SiO_2_ and ZrO_2_ or barium glass, respectively [32,33]. At the same time, BC is composed of a dental glass and a resin matrix with SiO_2_ and pigments [34]. This information suggests that LU and BC will be akin. However, after the microstructural and compositional analysis, BC and CS are quite similar, whereas LU differs slightly, which affects the mechanical properties.

The spectrum (Figure 4) and the element concentration (Table 2) were obtained from the XRF analysis for each material. All materials exhibit a broad peak around 22° which reveals an amorphous polymeric phase; besides, LU shows four characteristic peaks that match the ZrO2 (Figure 4). BC and CS do not display any crystalline indication. Likewise, BC and CS’s whole spectrum is very similar, which agrees with the obtained composition (Table 2). Results are in accordance with other authors [35].

The particle size distribution of the materials is different between them (Figure 5) and slightly coarser than reported by the manufacturers [36] but according to other authors [35]. At low magnification, two settings are observed. Whereas LU (Figure 5a) reveals a matrix with particles embedded with a size between 0.2–6 µm, BC and CS (Figure 5c,e, respectively) exhibit a particle size in the nanometre-scale: 0.1 µm to 2 µm for BC and 0.1 µm (or even smaller) to 0.5 µm for CS. However, beholding them at higher magnification, the particles’ morphology shares a likeness between LU and BC (Figure 5b,d). What is shared for the three materials is some holes between the grains and the matrix, although this pattern is more evident in the materials with the finer grain size (BC and CS).

### 3.2. Density

There are a few differences between the three materials’ experimental density with values around 1.9 g/cm^3^ with an outstanding high repeatedly (between 0.3 and 0.05%). It is interesting to note that this physical parameter is very relevant to estimate the possible porosity and proper compaction of materials that manufacturers do not usually reflect in their specification reports. Indeed, comparison data were only found in the case of LU with a value of 2.1 g/cm^3^ [36] and, although there is a 10% overestimated, values are comparable to those reported by other authors [37].

### 3.3. Micro and Nanohardness

Hardness values from Vickers and nanoindentation tests for LU are approximately 50% higher than those of BC and CS. Nevertheless, LU exhibits a more significant influence on the applied load and, therefore, higher dispersion of results (Figure 6a). This fact is due to its coarser microstructure and heterogeneous microstructure (Figure 5). At the nanoscale, results may vary as a function of the particular grain indention, which does not happen in BC and CS with all the results among their standard error.

In contrast, when larger loads are used, such as those used in Vickers indentation, they are averaged over larger areas. The value tends to include small porosities and material defects. This effect explains why the LU values with HV_AR for 9.8 N are 15% lower than the other methods’ results.

Although it was only possible to compare LU with the manufacturer’s data, since the information was not found in the literature for BC and CS, there is no clear or detailed information of how this value was measured and the employed conditions [32]. In the case of LU, 1 GPa hardness was obtained; other authors reported even lower values [38] for a load of 98 N, which reflects the vital influence of the applied load in materials with a coarse particle size distribution such as LU.

Otherwise, although the saliva immersion’s influence could seem residual since results are slightly overlapped with their standard error (Figure 6b), a clear tendency to significantly degrade the three materials can be observed. This tendency is entirely in line with the results obtained for other mechanical properties shown below.

### 3.4. Elastic Modulus

There are two techniques to determine the elastic modulus (E) used to compare our results with the MN and other authors’ ones. The first is the Impulse Excitation Technique (IET) that is very easy and quick to implement; however, when small dimension samples are used, the obtained results are quite erratic, is neither consistent nor reliable. For this reason, these results were not considered in our study, and the elastic modulus was calculated from the force-displacement curves obtained from the TPB tests of the smooth samples in both states AR and AS. For this latter method, a significant consideration to get the real elastic modulus is the correction of the stiffness of the whole loading system, which was corrected for each TPB strength tests.

Additionally, the elastic modulus was measured by nanoindentation tests because this is a relatively quick and precise method that requires a small amount of material. In this case, we propose to validate this method for future applications.

The results in Figure 7 show the elastic modulus obtained from nanoindentation and TPB tests and the data from the MN, which was reported after immersion in water for 24 h [34]. However, they do not explain in the literature why using this short period is insufficient to produce significant changes in the elastic modulus.

The higher elastic modulus values were obtained for the LU, then CB and finally CS in all cases. In general, MN provides values around 15–35% overestimated; however, a small disagreement was found in CS with various data depending on the source [34,39].

The direct comparison with elastic modulus obtained from other authors cannot be appropriately performed because of the diversity of conditions and methods employed for the measurements. For example, Belli et al. [37] reported an elastic modulus of 12.7 GPa for LU obtained through IET. As previously explained, it was not possible to use this method in our case. However, our results are similar to TPB values in the AR-state: 12.9 ± 1.4 GPa. Moreover, Tsujimoto et al. [40] reported values from TPB tests of 13.8 GPa for LU and 12.2 GPa for CS after immersion for 24 h in water for AS, while values obtained for AS immersion where 9.4 ± 0.6 GPa for LU and 7.1 ± 0.2 GPa for CS, 25% lower.

Five relevant outcomes can be established from our measurements:The nanoindentation method is a valid approximation to measure the elastic modulus of these materials. Nevertheless, this method tends to overestimate the modulus of elasticity, as it is a local measurement and does not take into account the defects and cracks present in the material. However, the latter weakness can also be attributed to the commonly used IET method since it is not sensitive to the presence of natural flaws.The EIT method shows significant reproducibility limitations for its use on such small samples, so its results should be considered exceptional because of its attention to its methodology.The ageing process’s influence exhibits a higher impact on the LU and CS and a few for CS following the rest of the mechanical properties.The best results of E are obtained for LU material, in both conditions AR and AS.The results of E measurements carried out through TPB tests (or tensile tests if possible) are more realistic than those obtained by other methods—such as nE or IET—since the material is characterized as a whole and the determination of E takes into account the defects and cracks present in the material.

### 3.5. Flexural Strength

The results obtained from the TPB tests (Figure 8) of materials tested in the AR-state result are very similar, with values around 150 MPa, 25% below expectations due to the values reported by the MN, which are around 200 MPa higher. For both conditions (AR and MN), flexural strength was measured by TPB tests, as explained in Section 2.6 and resulting in stress-strain linear curves until fracture. The information about the conditions in which the tests were performed was not specified in many cases by the providers. Additionally, various flexural strength values were found for the same material as a function of the source that provides them. The latter is the case for LU [32,33], exhibiting values over 25% or 30% and over 65% in the case of CS [33].

The degradation of the flexural strength because of the ageing process is also remarkable in LU and CS. This influence of the immersion in artificial saliva is more significant in the LU than in the other materials, with a detriment of 25%. It is the material with the coarser microstructure (Figure 5). The BC material does not present influence from the ageing process and, in comparison with the technical data, values reported after storage in water only for 24 h give values around 36% over the σ_AS_ [34], which may not be enough time to reach the saturation of the material [22].

### 3.6. Fracture Toughness

Results for the fracture toughness were obtained through TPB tests in SELNB specimens (Figure 9). The AR-state values for BC are higher than LU and CS, but all of them range from 1 MPa·m^1/2^. The results of LU—the only substance with previously reported data—are in agreement with Belli et al. [41], stating the fracture toughness at approximately half of MN’s value. The obtained values are below the dentin fracture toughness, 1.79 ± 0.06 MPa·m^1/2^, obtained under similar conditions—TPB tests with highly sharp samples [42]. This fracture toughness value of the dentin, which is larger but more similar than previously reported results [43], guarantees a similar behaviour between the natural teeth and the material and the protection of the natural teeth due to its lower brittleness.

The ageing process’s influence follows the same tendency of the values for flexural strength (Figure 8): a decrease of 30% for LU and half for BC and CS, around 15%.

### 3.7. Fracture Surfaces

Although the three materials have a resin matrix, they exhibit a brittle fracture behaviour. There is no evidence of ductility in analyzing the stress-strain curves from the TPB tests or the fracture surfaces’ morphology (Figure 10).

This result agrees with the macroscopic scale observations, where the surfaces of the three materials showed flat breakage on a plane perpendicular to the load without plastic deformation at the microscopic level.

## 4. Discussion

Given the considerable number of experimental techniques used and results obtained for the materials analyzed, as can be seen above, the results were discussed when they were presented. The aim was to achieve a better analysis and interpretation of the behaviour of the materials. To avoid repetitions, in this section, we will make an integrated analysis of all of them to understand the characteristics of each one of the materials globally.

An in-depth study has been conducted on the characteristics of three resin-based CAD/CAM composites used for indirect dental restorations. This study has analyzed the composition, microstructure, density and overall mechanical behaviour of the three materials, both as received and after 30 days in saliva (time needed to reach saturation of mechanical properties according to previous studies by other authors [22]).

The most remarkable fact of the results obtained for all the measurements made is that the three materials show consistent trends that allow us to state that immersion in saliva for 30 days significantly modifies the material’s mechanical behaviour in all the aspects analyzed. Reductions in properties of more than 30% (as is the case of fracture toughness) can be caused, which can seriously compromise the service materials’ long-term viability.

From the results obtained, it is significant that the material with a coarser microstructure and a greater presence of zirconia (LU) is the one that always presents the best mechanical parameters. On the other hand, although the three materials show gaps between the grains and the matrix, microstructural observations suggest that these are smaller in the LU, which is consistent with their higher overall mechanical strength. However, this compositional and microstructural strength is also its greatest weakness, as it results in the material degrading most intensely after immersion in saliva. This effect is probably due to the high sensitivity of zirconia to saliva, which produces its microstructural destabilization, going from a tetragonal phase to a monoclinic phase that is much more fragile.

## 5. Conclusions

The main conclusions of this study can be resumed as follow:Element concentration of the materials and their nanoparticle size slightly differs from the producer’s technical data supply, as shown by data obtained from the X-ray fluorescence and EDX spectrometer analysis. BC and CS achieved considerably finer microstructure than LU. It is also relevant to the different zirconia concentration, relatively higher in LU than in BC and CS.Mechanical performance of the three materials is profoundly affected by microstructure and composition. The best results are obtained for LU, probably due to the higher zirconia content and lower apparent porosity.After the ageing process under 30-days immersion in saliva, very relevant changes were observed in the analyzed materials’ global mechanical behaviour, which can be critical for clinical applications. As a general rule, the materials’ degradation with the finer microstructure (BC and CS) is lower than in LU. Future studies will be necessary for developing a greater depth in finding the relationship between these macroscopic changes and compositional and/or microstructural changes in the materials due to their prolonged immersion in saliva.Among the resin-based materials analyzed, although BC had a hardness around 0.8 GPa, 20% lower than LU, it is the material with the higher fracture toughness and, what is more important, the least affected by the saliva ageing. This result may represent an enhanced behaviour inside the mouth since it suffers a reduce detriment of the mechanical properties.The nanoindentation method is a valid approximation to measure the elastic modulus of these materials. Nevertheless, this method tends to overestimate the modulus of elasticity, as it is a local measurement and does not take into account the defects and cracks present in the material. However, the latter fault can also be attributed to the commonly used IET method since it is not sensitive to natural flaws.The IET method shows significant reproducibility limitations for its use on such small samples, so its results should be considered exceptional, paying attention to its methodology.The results of E measurements carried out through three-point bending tests (or tensile tests if possible) are more realistic than those obtained by other methods, such as nE or IET, since the material is characterized as a whole and the determination of E takes into account the defects and cracks present in the material.The information provided in manufacturers’ technical reports is often minimal, incomplete and poorly referenced. Therefore, independent studies are needed to obtain complete and objective information about the materials before being used in the clinic.

## Figures and Tables

**Figure 1 polymers-13-00808-f001:**
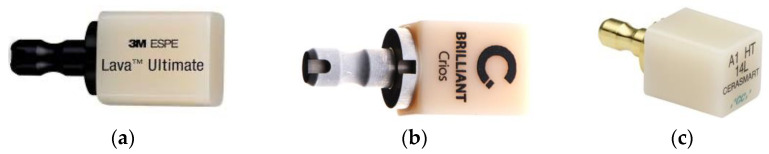
Chairside assistant computer-aid design and manufacturing technologies (CAD/CAM) C14 blocks: (**a**) Lava^TM^ Ultimate Restorative (LU), (**b**) Brilliant Crios (BC), (**c**) Cerasmart^TM^ (CS).

**Figure 2 polymers-13-00808-f002:**
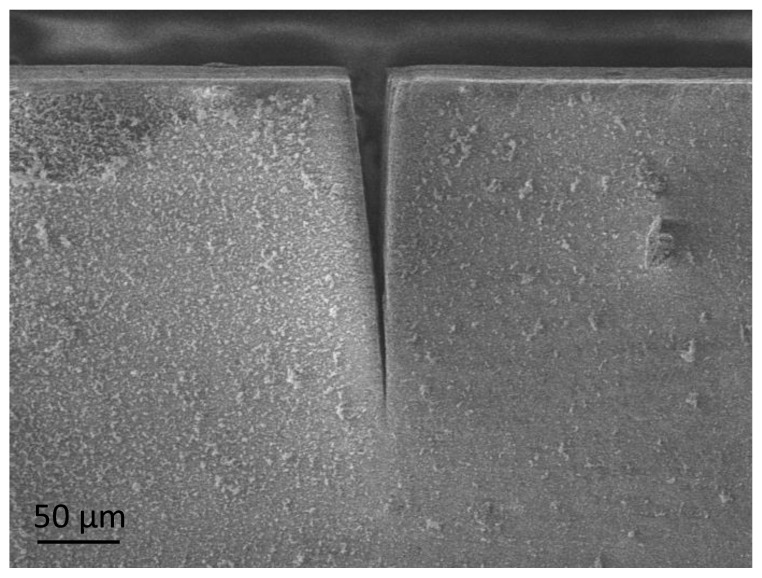
Field emission scanning electron microscopy (FESEM) image of the single edge laser-notch beam (SELNB) sample of LU.

**Figure 3 polymers-13-00808-f003:**
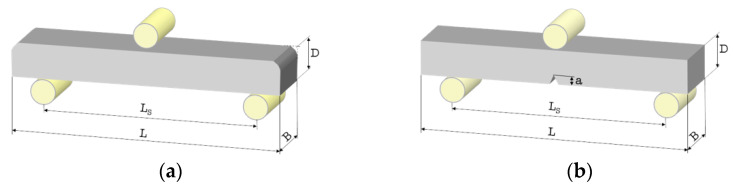
Three-point bending (TPB) device and (**a**) the smooth beam specimen for flexural strength measurements and (**b**) the SELNB specimen for fracture toughness measurements. L = 17 mm is the nominal sample length, B = 1.5 mm the sample width, D = 1.5 mm the sample height, L_s_ = 12 mm the load span and “a” the notch length, which was measured for each specimen.

**Figure 4 polymers-13-00808-f004:**
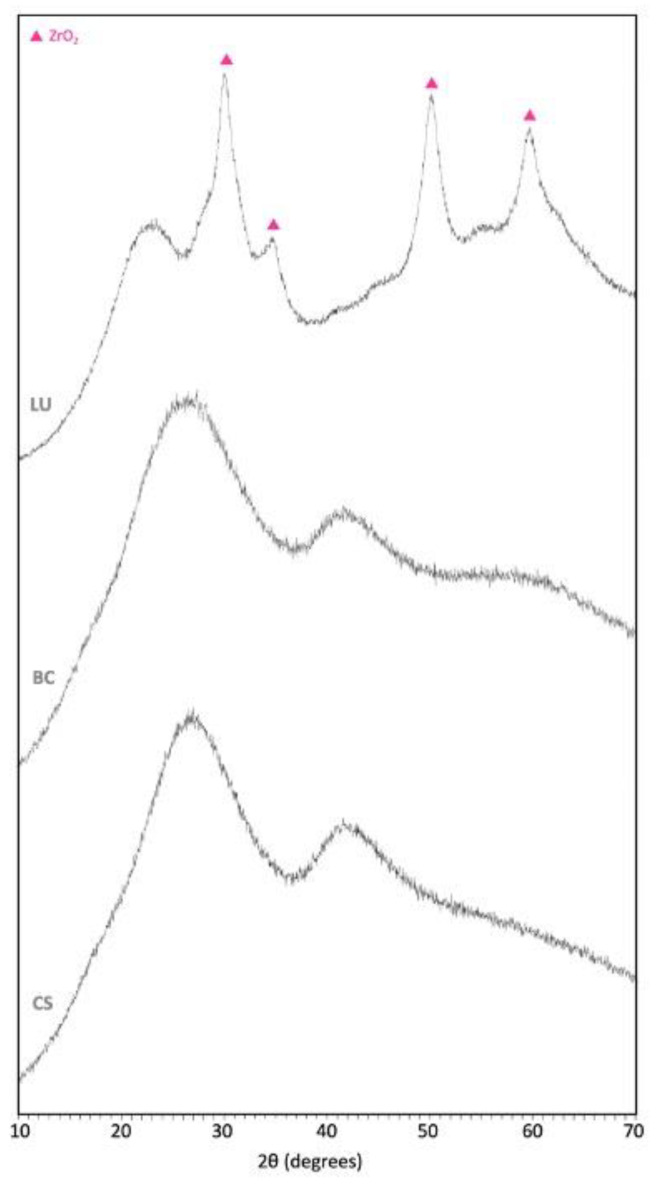
X-Ray fluorescence (XRF) spectrum of LU, BC, CS.

**Figure 5 polymers-13-00808-f005:**
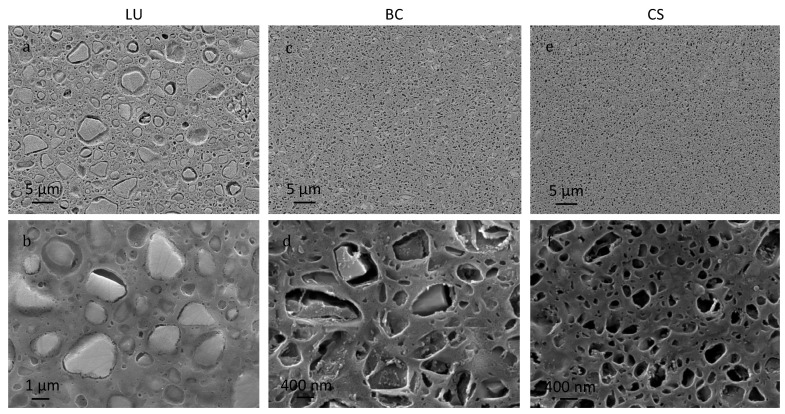
FESEM micrographs of polished and etched surfaces of (**a**,**b**) LU, (**c**,**d**) BC, and (**e**,**f**) CS materials. Note that micrography of LU has a lower magnification than BC and CS.

**Figure 6 polymers-13-00808-f006:**
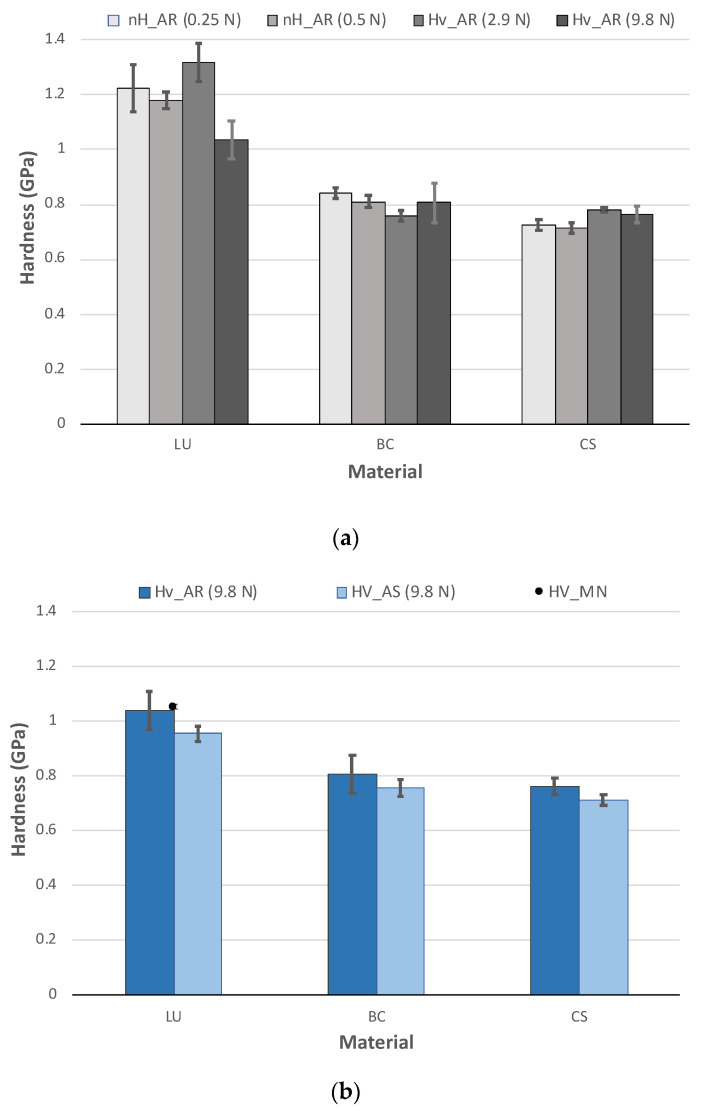
Micro and nanohardness (mean and standard error) (**a**) Influence of the applied load and the test method on the received from manufacturer (AR) specimens; (**b**) Influence of the artificial saliva (AS) immersion and the comparison with the manufacturer (MN) data.

**Figure 7 polymers-13-00808-f007:**
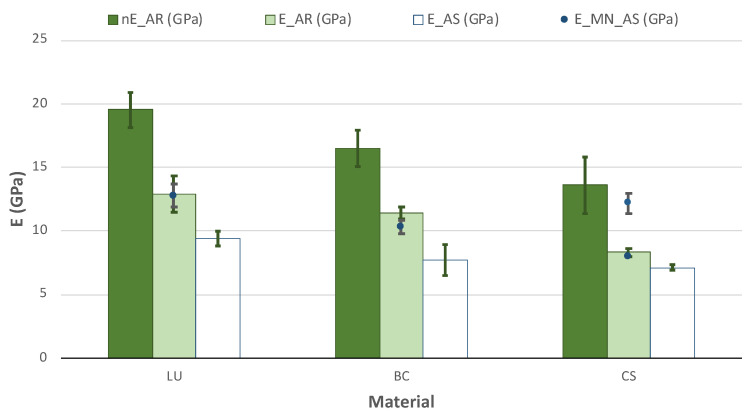
Nanoelastic modulus (nE) and elastic modulus (E) from TPB tests and standard error of the materials in the AR-state and after immersion in AS and compare with the MN’s data after water immersion.

**Figure 8 polymers-13-00808-f008:**
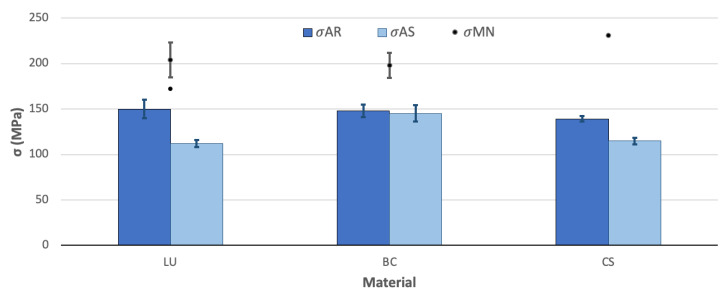
Mean flexural strength and standard error after the TPB tests on smooth samples for the materials AR and after immersion in AS and the comparison with the MN data (data for the material BC is reported in the literature after 24 h storage in water).

**Figure 9 polymers-13-00808-f009:**
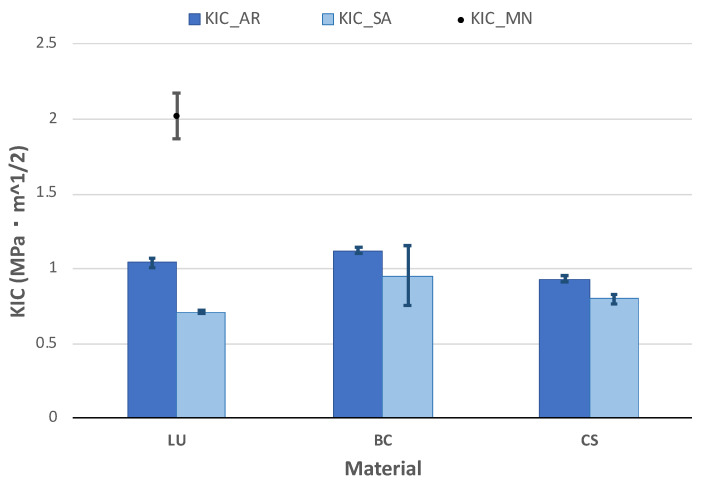
Mean fracture toughness and standard error after the TPB tests in the SELNB samples of the materials AR and after immersion in AS. Dentin fracture toughness is marked with the shaded area.

**Figure 10 polymers-13-00808-f010:**
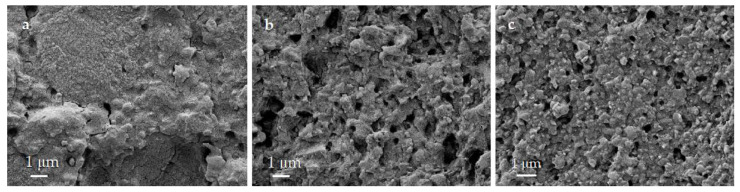
Fracture surfaces of the (**a**) LU, (**b**) BC (**b**), and (**c**) CS materials after a TPB test.

**Table 1 polymers-13-00808-t001:** Composition of the studied materials from their manufacturers’ technical documentation.

Material	Abbreviation	Manufacturer	Type
Lava^TM^ Ultimate	LU	3M ESPE, St. Paul, MN, USA	Composite resin material (UDMA, BisGMA, BisEMA, TEGDMA) with 80 wt.% SiO_2_ (20 nm) and ZrO_2_ (4–11 nm) nanoparticles and ZrO_2_/ SiO_2_ nanoclusters.
Brilliant Crios	BC	Coltene, Langenau, Germany	Barium glass (<1 µm), amorphous SiO_2_ (<20 nm), cross-linked methacrylates and inorganic pigments: ferrous oxide or titanium dioxide.
Cerasmart^TM^	CS	GC Dental Product, Tokyo, Japan	Composite resin material (UDMA, BisMEPP, DMA) with 71 wt. % SiO_2_ (20 nm) and barium glass (300 nm) nanoparticles.

UDMA: urethane dimethacrylate; BisGMA: bisphenol A diglycidyl ether dimethacrylate; BisEMA: ethoxylated bisphenol A dimethacrylate; TEGDMA: trimethylene glycol dimethacrylate; BisMEPP: 2,2-bis (4-methyacryloxy polyethoxy phenyl) propane; DMA: dodecyl dimethacrylate.

**Table 2 polymers-13-00808-t002:** Element concentration (%) obtained from the XRF analysis.

Material	SiO_2_	ZrO_2_	BaO	Al_2_O_3_	Na_2_O	F	HfO_2_	K_2_O	Other
LU	66.31	31.63	-	0.29	-	-	0.57	0.37	0.83
BC	62.43	2.58	23.91	9.02	0.69	-	-	0.13	1.24
CS	61.43	-	28.65	8.59	0.37	0.58	-	0.02	0.36

## Data Availability

Not applicable.

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
