# Peer review of "Saliva Influence on the Mechanical Properties of Advanced CAD/CAM Composites for Indirect Dental Restorations"

_polymers, 2021, doi:10.3390/polym13050808_

Round 1

Reviewer 1 Report

I have reviewed the manuscript “Saliva influence on the mechanical properties of advanced CAD/CAM composites for indirect dental restorations” submitted to “Polymers” for publication. In this study, authors have evaluated the microstructural and mechanical properties of three commercially available resin-based materials available for CAD/CAM-processed indirect dental restoration and characterized all the three types of resin-based composite CAD/CAM materials for physical and mechanical properties under various conditions.

This is a well-designed and well conducted study and the manuscript fits well within the scope of the journal; it needs some major improvements; there are a few suggestions that authors may consider to improve it further:

The use of English language is reasonable, however, there are a number of punctuation and grammatical errors; that should be corrected and rephrased using academic English for a better flow of text for reader.

Abstract: is very well presented and included all the key details. The conclusive statements (end the end) can be better present in the abstract.

Introduction: Is there any references to cite for the information presented in the Line 35-52: please cite appropriate references where possible to support these statements.  

Line 60-62: this is the key statement; however only one reference cited; authors are advised to add more citations; following papers can be included in addition to ref#5.

Influence of various specimen storage strategies on dental resin-based composite properties." Materials Technology (2021); 36(1): 54-62.

Effects of deformation rate variation on biaxial flexural properties of dental resin composites." Journal of Taibah University medical sciences 13.4 (2018): 319-326.

There is some issue with references and citations; as I could not see all the citations in sequence; please revise accordingly.

Methods and Results are well presented and detailed; there are some minor changes to improve.

Table 3 can be removed as this information can be presented in the text in a single line.

Figures 6-9: please bold/enlarge the fonts and error bars for further clarity

Figure 10: please add scale bars: also indicate Fracture surfaces using some markers

The discussion is very brief; almost no discussion; there are no citations in context: author should either expand discussion with citing related studies or merge it with the results section.

What are the limitation of this study? Should be included.

The conclusion section is too detailed; it is better to keep the conclusive remarks concise and some of the related information can be moved from conclusion section to the discussion section.

Author Response

Reviewer#1:

I have reviewed the manuscript "Saliva influence on the mechanical properties of advanced CAD/CAM composites for indirect dental restorations" submitted to "Polymers" for publication. In this study, authors have evaluated the microstructural and mechanical properties of three commercially available resin-based materials available for CAD/CAM-processed indirect dental restoration and characterized all the three types of resin-based composite CAD/CAM materials for physical and mechanical properties under various conditions.

This is a well-designed and well conducted study, and the manuscript fits well within the scope of the journal; it needs some major improvements; there are a few suggestions that authors may consider to improve it further:

The use of English language is reasonable, however, there are a number of punctuation and grammatical errors; that should be corrected and rephrased using academic English for a better flow of text for reader. English has been checked for a native speaker, and the global quality improved.

Abstract: is very well presented and included all the key details. The conclusive statements (end the end) can be better present in the abstract.2

Introduction: Is there any references to cite for the information presented in the Line 35-52: please cite appropriate references where possible to support these statements.  

The introduction was rewritten (lines 35-71), and references from 1 to 7 were included in the manuscript.

Line 60-62: this is the key statement; however only one reference cited; authors are advised to add more citations; following papers can be included in addition to ref#5.

  • Influence of various specimen storage strategies on dental resin-based composite properties." Materials Technology (2021); 36(1): 54-62.
  • Effects of deformation rate variation on biaxial flexural properties of dental resin composites." Journal of Taibah University medical sciences 13.4 (2018): 319-326.

References from 12 to 16 were included to support the statement in line 78-79.

There is some issue with references and citations; as I could not see all the citations in sequence; please revise accordingly.

References and citations have been revised appropriately with new inclusions.

Methods and Results are well presented and detailed; there are some minor changes to improve.

Table 3 can be removed as this information can be presented in the text in a single line.

Table 3 have been removed, and all the information is in the text now.

Figures 6-9: please bold/enlarge the fonts and error bars for further clarity

We have changed the format of the error bars for a clear understanding. However, some of them are overlapped, so is not going to see them clearly in any case.

Figure 10: please add scale bars: also indicate Fracture surfaces using some markers

Figure 10 already have scale bars.

The discussion is very brief; almost no discussion; there are no citations in context: author should either expand discussion with citing related studies or merge it with the results section.

Given the considerable number of experimental techniques used and results obtained for the materials analyzed, as can be seen, the results were discussed as they were presented. Our short discussion aims to give the reader a better analysis and interpretation of the behaviour of the materials. In any case, a paragraph has been included to clarify it.

What are the limitation of this study? Should be included.

This research is a fundamental study from scratch, so that there is not any limitation to it.

The conclusion section is too detailed; it is better to keep the conclusive remarks concise and some of the related information can be moved from conclusion section to the discussion section.

Given the depth and extent of the study carried out, the fact that we worked with three materials in two different conditions for each one and the diversity of results obtained, it is impossible to abbreviate the conclusions without depriving the reader of vital information to understand the study carried out in its entirety.

Reviewer 2 Report

Thanks for submitting this manuscript.

I have carefully read your manuscript with great interest.

I think that it should sound very interesting the change of material properties by saliva for readers and this paper overall well written.

However, the authors are missing the most important point.

It could be supporting the value of study by acquired data reliability on statistical analysis.

All experimental data did not show the number of experiments including independent repeated experiments.

Authors must be clear this point. Then, authors need to address these statistical issues (parametric or non-parametric test, normality, one-way ANOVA etc) in manuscript.

Authors will be able to re-analyze the results by appropriate statistical analysis, after that, it may happen that the results of some statistical comparisons may change and consequently the discussion.

Sincerely,

Author Response

Thanks for submitting this manuscript.

I have carefully read your manuscript with great interest.

I think that it should sound very interesting the change of material properties by saliva for readers and this paper overall well written.

However, the authors are missing the most important point.

It could be supporting the value of study by acquired data reliability on statistical analysis.

All experimental data did not show the number of experiments including independent repeated experiments.

Authors must be clear this point. Then, authors need to address these statistical issues (parametric or non-parametric test, normality, one-way ANOVA etc) in manuscript.

Authors will be able to re-analyze the results by appropriate statistical analysis, after that, it may happen that the results of some statistical comparisons may change and consequently the discussion.

Throughout this work, a statistical study of the results obtained has been carried out based on the Gaussian statistical distribution that usually presents materials' mechanical and physical characterization. It can be seen that for each measured parameter, the interval in which the actual value is given with a 68 % probability, based on the mean value and the bandwidth defined by the mean square error.

This statistic assumes that all materials are intrinsically equal, which is true because they have all been produced in the same way and undergone the same treatments. Under these conditions, it is always to be expected that the results will follow a repeating distribution that fits the Gaussian bell.

Using other types of statistics, such as those proposed, does not provide more information to the study. The reviewer's statistical analyses are appropriate for populations where each sample or individual differs from the others. For example, when analyzing the effect of a treatment on a patient. Since this is not our case, the use of such statistical analysis, which we have performed but not presented, does not significantly contribute to the research. Moreover, showing the two types of statistical analysis may confuse a reader unfamiliar with using different statistical procedures depending on the characteristics of the populations analyzed.

Round 2

Reviewer 1 Report

Dear Authors, thank you for responding to comments. 

Reviewer 2 Report

Thanks for submitting revision.

Authors stated “a statistical study of the results obtained has been carried out based on the Gaussian statistical distribution that usually presents materials' mechanical and physical characterization.”

I agree this point, but that’s state present the distribution of data.

I just wonder that “How to analysis the comparison for mechanical properties before/after saliva immersion.”

In abstract, authors stated "these values were significantly reduced after one month of immersion in saliva."

The significant reduction is supported by statistical analysis.

Therefore, Authors must to add the number of experiments including independent repeated experiments.

Authors must be clear this point.

Authors will be able to re-analyze the results by appropriate statistical analysis for comparison, after that, it may happen that the results of some statistical comparisons may change and consequently the discussion.

Sincerely,